# Radiomic Consensus Clustering in Glioblastoma and Association with Gene Expression Profiles

**DOI:** 10.3390/cancers16244256

**Published:** 2024-12-21

**Authors:** Tadeusz H. Wroblewski, Mert Karabacak, Carina Seah, Raymund L. Yong, Konstantinos Margetis

**Affiliations:** 1College of Medicine, SUNY Downstate Health Sciences University, Brooklyn, NY 11203, USA; tadeusz.wroblewski@downstate.edu; 2MD-PhD Program, SUNY Downstate Health Sciences University, Brooklyn, NY 11203, USA; 3Department of Neurosurgery, Mount Sinai Health System, New York, NY 10029, USA; mert.karabacak@mountsinai.org (M.K.); raymund.yong@mountsinai.org (R.L.Y.); 4Department of Genetics and Genomic Sciences, Icahn School of Medicine at Mount Sinai, New York, NY 10029, USA; carina.seah@icahn.mssm.edu

**Keywords:** glioblastoma, radiomics, gene expression, clustering, unsupervised machine learning

## Abstract

Glioblastoma (GBM) is an aggressive primary central nervous system tumor with poor survival outcomes and limited treatment options. In this study, we investigate the use of radiomic features derived from magnetic resonance imaging (MRI) scans to identify unique gene expression profiles in a cohort of patients with GBM. This study grouped patients based on radiomic features using a consensus clustering approach, which iteratively clusters patients to find robust and stable groups. We identified three clusters which yielded unique gene expression profiles. Significant differentially expressed genes previously associated with GBM prognosis and treatment sensitivity were identified in one cluster. In pathway enrichment analyses, genes upregulated in immune-related and DNA metabolism pathways and downregulated protein and histone deacetylation pathways were identified in the same cluster. Together, these findings suggest that consensus clustering of radiomic features may be a promising avenue for non-invasive characterization of molecular heterogeneity of GBM.

## 1. Introduction

Glioblastoma (GBM) is the most common malignant primary central nervous system tumor, with an incidence of 3.19 per 100,000 in the United States [1]. The prognosis for patients with GBM remains poor, with a median overall survival of only 12–15 months [2,3,4]. Molecular profiling has emerged as a crucial factor in establishing GBM prognosis, as evidenced by the 2021 revised World Health Organization (WHO) tumor classification [5]. This updated classification has further integrated molecular data for typing, subtyping, and grading GBM, building upon the 2016 edition [6]. Despite these advancements, GBM continues to exhibit diverse tumor microenvironments and heterogeneous disease progression, posing significant challenges for characterization [7,8]. Consequently, there is a pressing need for further investigation into alternative techniques to classify these tumors and predict their clinical course more accurately.

Deciphering the complex molecular landscape and variability across GBM typically requires invasive tumor tissue sampling, often only possible during surgical intervention. Radiomics analysis offers a non-invasive alternative to gain insights into disease progression and tumor biology [9,10]. Magnetic resonance imaging-based (MRI) radiomics involves the computational extraction and quantification of sub-visual radiographic features that may reflect underlying biological characteristics and correlate with phenotypic traits, enabling the differentiation of intratumor heterogeneity [11,12]. Genomic alterations, frequently associated with GBM progression, have been correlated with radiomic feature analyses [12,13,14,15]. This association positions radiomics as a promising surrogate for classifying molecular differences between tumors.

The classification of radiomic features necessitates clustering algorithms to identify similarities and differences across datasets. Consensus clustering emerges as a promising approach to achieve this goal [16]. Unlike traditional clustering methods, which are prone to overfitting and inconsistency due to random initialization, consensus clustering yields stable and reproducible groups through multiple iterations of clustering. This iterative process assesses the stability of the resulting groups, enabling the identification of robust clusters. These clusters can then be visually evaluated to reveal biologically meaningful differences that correlate with molecular signatures across tumors.

We hypothesize that consensus clustering of radiomic features will offer a non-invasive method to identify molecularly relevant GBM groups, which can be further characterized through gene expression analysis. Furthermore, we posit that the identified clusters may reveal novel GBM subtypes with distinct gene expression signatures. In this study, we apply unsupervised consensus clustering to radiomic features extracted from MRI sequences of GBM patients to investigate the association between radiomic-based clusters and gene expression profiles. Our aim is to demonstrate that this consensus-clustering approach to radiomic features can effectively define distinct GBM groups with unique gene expression profiles.

## 2. Materials and Methods

### 2.1. Glioblastoma Cohort and Data Source

This study utilized data from three publicly available datasets: The Cancer Genome Atlas (TCGA) PanCancer Atlas (https://gdc.cancer.gov/about-data/publications/pancanatlas accessed on 7 August 2024) glioblastoma (TCGA-GBM) dataset [17], brain lower grade glioma (TCGA-LGG) dataset [18], and the Clinical Proteomic Tumor Analysis Consortium (CPTAC) GBM (CPTAC-GBM) dataset [19]. The TCGA-GBM and TCGA-LGG datasets comprise 617 patients with GBM and 516 patients with a historical histopathological diagnosis of lower grade glioma, respectively. Tumor grades for samples in the TCGA datasets were reassigned based on the recent reclassification of diffuse glioma profiles [20], in accordance with the 2021 WHO CNS tumor classification criteria [5]. Consequently, tumors previously diagnosed as lower grade in the TCGA-LGG dataset were included if reclassified as GBM. The CPTAC-GBM dataset consists of 200 patients with GBM. All three datasets contain imaging data, molecular data including gene expression profiles, and clinical information for a subset of patients across studies.

Inclusion criteria for this study were a GBM diagnosis based on the 2021 WHO grading criteria, along with the availability of both multi-parametric MRI (mpMRI) data and clinical information. Patients without a GBM diagnosis were excluded from the study. The final cohort comprised 114 patients across the three datasets with available imaging data and clinical information: 73 patients from TCGA-GBM, 18 patients from TCGA-LGG (reclassified as GBM), and 23 from the CPTAC-GBM dataset. This study was limited to adult patients with GBM. Imaging data for all three datasets were accessed through The Cancer Imaging Archive [21]. Clinical information was retrieved from the National Institutes of Health (NIH) National Cancer Institute (NCI) Genomic Data Commons (GDC; https://portal.gdc.cancer.gov/ accessed on 7 August 2024).

### 2.2. Radiomic Feature Extraction

We analyzed each patient’s multi-parametric MRI (mpMRI) sequences, including T1-weighted (T1), T1 with contrast (T1c), T2-weighted (T2), and Fluid Attenuated Inversion Recovery (FLAIR), using PyRadiomics (version 3.0.1) to extract an extensive set of radiomic features [22]. For each patient, we generated three binary segmentation masks: necrotic core (NC), tumor core (TC), and whole tumor (WT). The NC mask delineated the necrotic and non-enhancing tumor core, the TC mask encompassed the enhancing tumor and necrotic core, and the WT mask included the entire tumor region with peritumoral edema. Our extraction process yielded a comprehensive set of radiomic features for each patient, derived from each MRI sequence (T1, T1c, T2, FLAIR) in combination with each of the three tumor masks (NC, TC, WT). To ensure consistency across all images, we employed a predefined parameter file for feature extraction.

The extracted features were organized into several distinct categories: first-order statistics, Shape-based features (3D), Gray-Level Co-Occurrence Matrix (GLCM), Gray-Level Run Length Matrix (GLRLM), Gray-Level Size Zone Matrix (GLSZM), Neighboring Gray Tone Difference Matrix (NGTDM), and Gray-Level Dependency Matrix (GLDM). For each feature category, we generated feature matrices capturing various image characteristics, including intensity patterns, textures, and shapes. An overview of the radiomic feature extraction process is displayed in Figure 1; a comprehensive introduction to radiomic feature extraction and its utility for brain tumor research is available elsewhere [12].

### 2.3. Clustering

A total of 14,616 radiomic features were initially extracted from the imaging data. To reduce multicollinearity, we implemented a two-step feature selection process. First, features with no variance (variance = 0) were eliminated (n = 16). Subsequently, highly correlated features were removed using a pairwise absolute correlation cutoff of 0.95, resulting in the exclusion of 9030 features. Importantly, additional sensitivity analyses demonstrate that radiomic cluster assignment remained strongly correlated (>0.9) when assessing different correlation cutoffs (i.e., 0.9, 0.95, 0.99, 0.999; Appendix A). Consensus clustering was then performed on the remaining filtered quantitative feature set (n = 5570) to categorize tumors into distinct groups based on their radiomic profiles.

Consensus clustering is a data-driven approach used to identify the optimal number of clusters (*k*) and determine cluster membership [13,23]. Cluster agreement is assessed for a subsample of data across multiple iterations to identify the most stable and most appropriate number of clusters. In this study, clustering of radiomic features was performed with the ConsensusClusterPlus (v1.66.0) package [24] across all patients with available imaging data (n = 114). For a number of clusters between 2 and 6, we iteratively subsampled the dataset 1000 times; for each iteration, 80% of patients (pItem = 0.8) and all features (pFeature = 1.0) were clustered using a k-means base algorithm and Euclidean distance calculation. For each iteration, 80% of participants are subsampled and their radiomic features are clustered using a k-means base algorithm; the remaining 20% of participants are sampled in other iterations such that each participant is equally represented across the 1000 iterations.

Stated another way, the k-means algorithm represents the clustering algorithm within the ConsensusClusterPlus framework for each iteration of consensus clustering; however, any clustering algorithm can be utilized. Unlike other unsupervised clustering approaches, which often choose an arbitrary number of clusters, the consensus clustering approach calculates the frequency with each patient pair clustered together across clustering iterations in order to determine an appropriate *k* that includes stable and well-defined clusters. The patient pair frequencies are displayed as the relative consensus matrix (Appendix A) and using the consensus matrix distribution, we can then compute the change in area under the cumulative distribution function (CDF). The CDF plot represents how well clusters for each *k* are separated, with the optimal number of clusters determined based on a remarkable decrease in delta area on the CDF plot, representing the *k* at which adding more clusters would not considerably improve group separation.

### 2.4. Gene Expression Analysis

Gene expression data from primary GBM tumor samples were available for 69 out of the 114 patients included in this study across the three datasets: TCGA-GBM (n = 28), TCGA-LGG (n = 18), and CPTAC-GBM (n = 23). Raw count data were downloaded from the NIH NCI GDC portal (accessed 7 August 2024). Gene expression data were only assessed from primary GBM, and expression data from recurrent tumors were excluded. Probes were annotated using the AnnotationDbi (v1.64.1) package with org.Hs.eg.db (v3.18.0). Genes for which there was no mappable Entrez Gene ID were removed (excluded n = 23,467; 39.5%) and genes were filtered to exclude low-expressed genes that did not meet the threshold of at least 20 counts in at least approximately 20% of samples (final probe n = 20,291).

Merging count data from three separate datasets introduced variance due to batch effects, which were corrected using negative binomial regression with the ComBat-seq function in the sva (v3.50.0) package [25]. For batch correction, the dataset source (i.e., TCGA-GBM, TCGA-LGG, and CPTAC-GBM) was assigned as the “batch” variable, and variance due to radiomic clusters, sex, and age was preserved. Race and ethnicity variables were strongly correlated with dataset source (Appendix A), demonstrated by higher counts of “unknown” and “other” responses in clinical data from CPTAC compared to TCGA; therefore, these variables were not considered as covariates for downstream analyses. Variance due to batch effects was examined with unsupervised Principal Component Analyses (PCAs), and confounding variables were examined using VariancePartition (v1.32.5) [26] before and after batch correction (Appendix A). After correcting raw count data for batch effects, adjusted expression values were converted to log_2_ (counts per million) and normalized using edgeR (v4.0.16) and limma (v3.58.1) [27] packages. Differentially expressed genes (DEGs) were identified by fitting a linear model (adjusted for patient sex and age) using weighted least squares and empirical smoothing of standard errors with limma. Nominally significant genes were identified by *p* < 0.01 and absolute log_2_ fold change > 0.5. Significant DEGs were established based on *p*-value adjusted < 0.05 and absolute log_2_ fold change > 0.5 after correction for multiple testing using Benjamini–Hochberg false discovery adjustment.

### 2.5. Gene Set Enrichment Analysis

Gene set enrichment analysis (GSEA) was performed on genes pre-ranked from DEG analysis based on statistical significance and direction of effects: −*log*_10_(*p*) × *sign*(*fold change*). GSEA was performed with fsgea (v1.28.0) [28] using the Gene Ontology (GO) biological processes gene set [29]. Significantly enriched pathways were established based on *p* < 0.05 after Benjamini–Hochberg false discovery correction.

### 2.6. Statistical Analyses

Demographic variables and clinical characteristics were compared across radiomic clusters using Kruskal–Wallis rank sum test for continuous variables and χ^2^ test or Fisher’s Exact test, when appropriate, for categorical variables. Continuous variables were displayed as the mean ± standard deviation (SD) and categorical variables were displayed as the sample n (percent, %). Overall survival (OS) was assessed with Kaplan–Meier curves with survival time considered as “days to death” for deceased patients and “days to last follow-up” for patients reported living. Differences in overall survival curves across radiomic clusters was assessed using log-rank test [30]. Median OS was calculated based on the Kaplan–Meier curves and displayed as median [95% confidence interval]. Significance was defined as *p* < 0.05. All statistical analyses were performed in R Statistical Suite (version 4.3.1).

## 3. Results

One-hundred-and-fourteen patients with GBM were included in this study and grouped into three clusters based on unsupervised consensus clustering of radiomic features; 25 patients were in Cluster 1, 46 were in Cluster 2, and 43 were in Cluster 3 (Table 1). The average patient age at the time of diagnosis was 60 ± 11 SD years, with no significant difference in age identified across clusters (*p* = 0.6, Table 1). There were more male patients in the cohort (60%; n = 68) compared to female (40%; n = 46), and the majority of participants were of White racial identity (83%; n = 94) and non-Hispanic or Latino ethnicity (73%; n = 83). No significant differences in patient sex (*p* = 0.6), racial identity (*p* = 0.2), or ethnicity (*p* > 0.9) were identified across radiomic clusters (Table 1). No significant difference MGMT promotor methylation status was identified across radiomic clusters amongst samples with available data (Table 1); however, a lower proportion of MGMT methylated tumors were identified in Cluster 1 (20%; n = 2) compared to Cluster 2 (52%; n = 16) and Cluster 3 (52%; n = 14). Amongst samples with available tumor location information (n = 41), there was no significant difference across clusters (Table 1). The median OS of the cohort was 14.1 [11.7–16.7] months, though no significant difference in overall survival was identified across clusters (*p* = 0.82; Table 1 and Appendix A).

A total of 69 patients across the three datasets had available gene expression data. DEGs were investigated across patients stratified into the three radiomic feature-based clusters (Cluster 1, n = 19; Cluster 2, n = 18; Cluster 3, n = 32). In total, 598 genes (255 upregulated and 343 downregulated) were considered nominally differentially expressed (*p* < 0.05) across the three clusters (Figure 2). Clusters 1 and 2 exhibited a predominance of downregulated genes compared to upregulated genes among all DEGs reaching nominal significance, with ratios of 1.29:1 and 1.75:1, respectively. In contrast, Cluster 3 displayed a nearly equal number of up- and downregulated genes (ratio 0.97:1). Notably, Cluster 1 was the only group to contain DEGs that remained significant after correction for multiple testing, with 9 upregulated (Table 2) and 19 downregulated (Table 3) genes identified in Cluster 1 that were significantly enriched, and 385 genes (168 upregulated and 217 downregulated) were nominally significant. No significant DEGs were identified in Clusters 2 and 3 after correction for multiple testing. However, 110 DEGs (40 upregulated and 70 downregulated) in Cluster 2 were nominally significant, and 75 genes (38 upregulated and 37 downregulated) that reached nominal significance were identified in Cluster 3. Furthermore, 42 genes were identified that achieved nominal significance in more than 1 cluster, including four genes nominally significant in Cluster 2 or 3 that were also significantly enriched in Cluster 1 after correction for multiple testing (Appendix A).

Gene set enrichment analysis (GSEA) was then performed for each cluster based on a pre-ranked gene list according to *p*-value and direction of change (i.e., upregulated or downregulated, Figure 3). Genes upregulated in Cluster 1 (i.e., genes at the top of the ranked gene list) were significantly associated with mast cell activation (*p* = 0.044) and increased adaptive immune response (*p* = 0.044) pathways, with normalized enrichment scores (NES) of 2.08 and 1.51, respectively. Additionally, pathways associated with upregulated genes in Cluster 1 included isoprenoid catabolic processing (NES = 2.06; *p* = 0.049); metabolic processes related to pyrimidine dNTPs (NES = 2.01; *p* = 0.022) and dTTP metabolism (NES = 1.88; *p* = 0.035); and androgen receptor regulation (NES = 1.86; *p* = 0.044). Genes downregulated in Cluster 1 (i.e., genes at the bottom of the ranked gene list) were enriched for positive regulation of protein and histone deacetylation (NES = −2.26; *p* = 0.018 and NES = −2.17; *p* = 0.044, respectively); and synaptic signaling (NES = −1.42; *p* = 0.036, Appendix A). No significantly enriched pathways were identified for ranked gene lists from Cluster 2 or Cluster 3 (Appendix A).

## 4. Discussion

Few studies have combined gene expression analysis with comprehensive radiomic-based clustering to identify unique molecular signatures in GBM. Our study aimed to assess the association between gene expression profiles and consensus clusters established using MRI-based radiomic features in GBM patients. Analysis of all patients revealed no significant differences in age, sex, race, or ethnicity across the three identified clusters. This finding suggests that the variations in radiomic features driving the clustering are likely attributable to subtle molecular differences rather than the demographic and epidemiological factors traditionally associated with GBM [4,31]. In other words, the variations in radiomic features likely represent subtle differences in tumor biology and radiosensitivity that are not visually discernible [10], underscoring the potential of radiomics to capture molecular heterogeneity in GBM. Together, these findings suggest that the heterogeneity captured by radiomic consensus clustering may represent molecular differences across GBM that are independent of patient demographics but may hold promise in the future for targeted therapies.

The underlying characteristics captured by the radiomic consensus clustering likely stem from alterations in the tumor’s molecular composition, including genetic, transcriptomic, and proteomic changes, which manifest as subtle differences detectable only through radiomic feature extraction. No difference in overall survival was identified across radiomic clusters in our cohort. The lack of association between patient survival and radiomic clusters may represent the significant intratumor molecular heterogeneity amongst GBM that contribute to unique molecular profiles which are not being selectively targeted with currently available treatment options [32]. That is to say, without targeted therapies for GBM, it is unlikely that survival would correspond to molecular profile. Consequently, analyzing the molecular signatures associated with different radiomic clusters may offer deeper insights into intratumor molecular heterogeneity, rather than merely reflecting differences across patient demographics or overall survival. Notwithstanding, our consensus clustering approach yielded distinct expression profiles across the three groups, aligning with previous findings [33,34].

While variation in overall survival was not associated with radiomic clusters broadly, genes differentially expressed across cluster have been previously shown to be associated with GBM pathogenesis and disease progression. Genes upregulated in Cluster 1 which have been previously associated with GBM progression include *BUD31*, *TRIP4*, *RPL39*, *C5orf46*, and *TFB2M*. Increased expression of *BUD31* [35], *RIP4* [36], and *RPL39* [37,38] have been previously correlated with disease malignancy and associated worse overall survival in patients with GBM, with in vitro overexpression of *RPL39* associated with enhanced proliferation and migration of tumor cells [37]. Upregulation of *C5orf46* has been implicated as a driver in GBM recurrence status and has been associated with poor prognosis in recurrent GBM [39].

Similarly, downregulated genes identified in Cluster 1 have been previously associated with more aggressive tumor characteristics and higher tumor grade; these include *RNF39*, *ADAMTS8*, *SLC25A42*, *ST3GAL5*, and *ZBED3*. Decreased expression of *RNF39* has been associated with lower survival probability in patients with glioma, with expression levels negatively correlated with WHO glioma tumor grade [40]. Furthermore, *ADAMTS8* [41,42] and *SLC25A42* [43] encode for proteins with strong antiangiogenic effects and antiproliferative properties, respectively, with decreased expression associated with increased tumor viability and proliferation. *ST3GAL5* is an inducer of apoptosis through the indirect suppression of EGFR/PI_3_K signaling pathways suggesting that *ST3GAL5* downregulation may contribute to increased cell proliferation [44]. Finally, *ZBED3* was also downregulated in Cluster 1, which has been previously associated with temozolomide resistance in GBM [45]. Interestingly, *DLG5* and *PRX* were also downregulated in Cluster 1, which have been previously associated with improved prognosis and favorable chemotherapy response. *DLG5* is associated with GBM stem cell invasion with under expression (demonstrated through knock down experiment) previously shown to cause disruption of GBM tumor formation and been associated with increased patient survival [46]. Furthermore, downregulation of *PRX* has been associated with better response to chemotherapy in glioma [47,48]. Despite promising evidence of DEGs identified in Cluster 1, no DEGs were identified in Clusters 2 and 3 that remained significant after correction for multiple testing, suggesting that a larger sample size may be required to further understand the molecular differences across radiomic clusters.

We then used gene enrichment analyses to further classify the biological pathways and cellular functions of DEGs in Cluster 1. Genes upregulated in Cluster 1 were primarily associated with immunological and metabolic pathways. Gliomas have been previously associated with mast cell accumulation and pro-inflammatory pathways that modulate angiogenic and immunogenic processes impacting the GBM microenvironment [49]. Positive enrichment for genes involved in mast cell activation and increased adaptive immune response suggests the potential role of immune activation as a contributor to the unique radiomic feature profile observed in Cluster 1. Furthermore, upregulation of genes involved in DNA metabolism pathways were identified in Cluster 1, suggesting increased drivers of cell proliferation are present in these tumors. Interestingly, downregulated genes in Cluster 1 were enriched for pathways associated with protein and histone deacetylation regulation. Histone deacetylation is a well-established epigenetic modification that occurs in GBM [50] with histone deacetylases (HDAC) implicated as a potential therapeutic target for GBM treatment [51]. In this study, we identify the differential expression of genes involved in histone deacetylation regulation across clusters (downregulated in Cluster 1), suggesting that clustering by radiomic features may hold utility in predicting underlying sensitivity to chemotherapies (e.g., HDAC inhibitors) across clusters.

We acknowledge several limitations of this study: we present a limited sample size warranting future large scale clustering applications that may provide useful insights into unique gene expression profiles that can drive targeted treatment based on molecular findings and enable appropriately powered sub-group analyses. Additional limitations of our study include the use of expression data from GBMs at a single time point in the disease progression. Radiomics provides a useful tool to non-invasively correlate the molecular status of GBM and further investigation is required across multiple time points to assess how this approach may capture changes over time. Finally, this study utilized gene expression data captured from multiple datasets, which poses an inherent risk of batch effects; in this study, these effects were addressed using a correction method specifically tailored for RNA-seq data.

## 5. Conclusions

Together, these findings demonstrate the potential utility of consensus clustering to differentiate GBM based on gene expression profiles which may inform therapeutic targets in the future. More specifically, these findings suggest that clustering by radiomic features can identify tumor groups with unique gene expression profiles that may impact cell proliferation and altered tumor morphology. Non-invasively identifying unique molecular profiles across a cohort of patients with GBM may provide further insight into the mechanisms of drug-resistance in tumors and provide additional information for selecting targeted therapies for GBM. Further investigation with a larger cohort of patients including tumors with different disease statuses is warranted in the future.

## Figures and Tables

**Figure 1 cancers-16-04256-f001:**
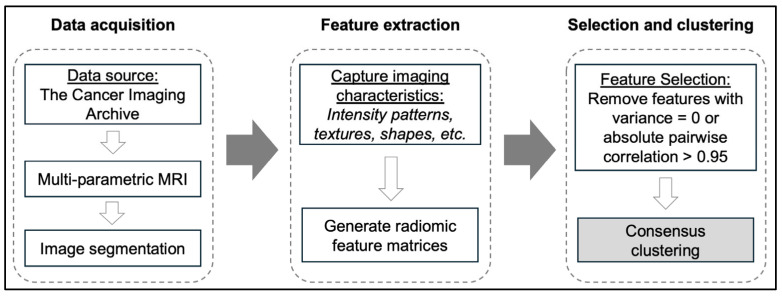
Diagram of radiomic feature extraction workflow. MRI = Magnetic resonance imaging.

**Figure 2 cancers-16-04256-f002:**
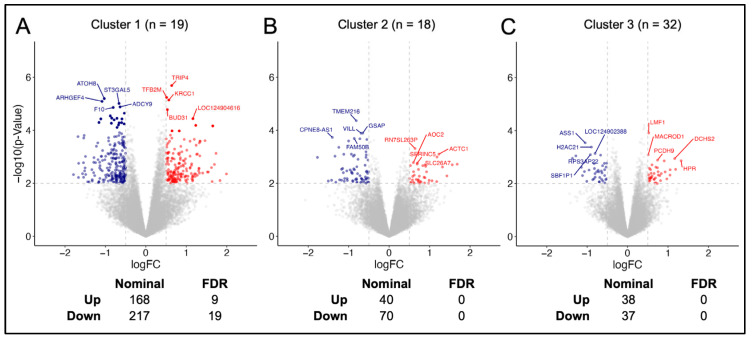
Differentially expressed genes (DEG) across radiomic consensus clusters. Volcano plots representing DEGs for (**A**) Cluster 1, (**B**) Cluster 2, and (**C**) Cluster 3. Horizontal dashed line represents *p* = 0.01 and vertical line represents absolute logFC = 0.5. Nominally significant represents un-adjusted *p* < 0.01 with absolute logFC > 0.5. Points representing nominally significant genes that are upregulated in red, and downregulated in blue. Significance determined as *p*-adjusted < 0.05 (FDR) with logFC (absolute log_2_ fold change) > 0.5.

**Figure 3 cancers-16-04256-f003:**
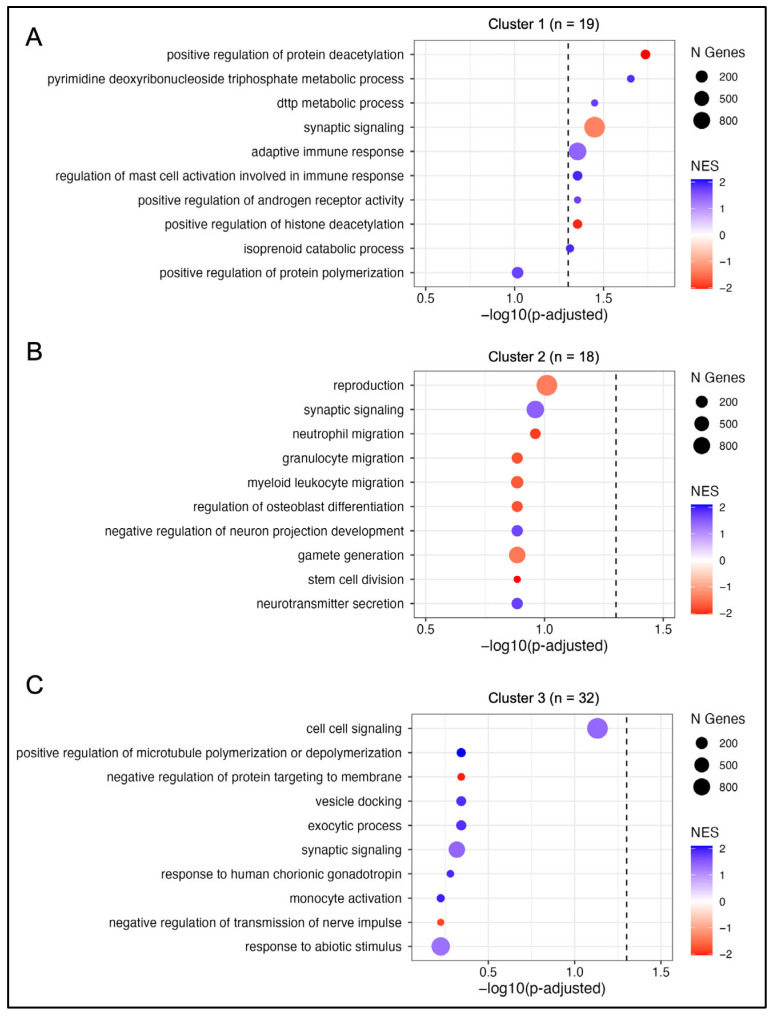
Upregulated pathways associated with differentially expressed genes in GBM. Top Gene Ontology biological process pathways for each radiomic consensus cluster (**A**–**C**). Size of each point indicates the number of genes (N Genes) represented in each gene set and color represents the normalized enrichment score (NES). The dashed vertical line represents *p*-adjusted = 0.05.

**Table 1 cancers-16-04256-t001:** Demographic and clinical characteristics for patients in the GBM cohort.

	Full Cohort ^1^	Cluster 1	Cluster 2	Cluster 3	*p*-Value ^2^
**Count**	114	25	46	43	-
**Age**	60 ± 11	62 ± 11	61 ± 12	59 ± 10	0.5
**Sex**	114				0.6
Female	46 (40%)	8 (32%)	19 (41%)	19 (44%)	
Male	68 (60%)	17 (68%)	27 (59%)	24 (56%)	
**Race**	103				0.2
Black/African American	9 (8.0%)	1 (5.6%)	2 (4.5%)	6 (15%)	
White	94 (83%)	17 (94%)	42 (95%)	35 (85%)	
Other or Unknown	10 (8.8%)	7	2	2	
**Ethnicity**	85				>0.9
Hispanic or Latino	2 (1.8%)	15 (100%)	34 (97%)	34 (97%)	
Not Hispanic or Latino	83 (73%)	0 (0%)	1 (2.9%)	1 (2.9%)	
Unknown	29 (25%)	10	11	8	
**MGMT Promotor Status**	68				0.2
Unmethylated	36 (32%)	8 (80%)	15 (48%)	13 (48%)	
Methylated	32 (28%)	2 (20%)	16 (52%)	14 (52%)	
Unknown or not measured	46 (40%)	15	15	16	
**Tumor Location**	41				0.8
Frontal Lobe	21 (18%)	4 (36%)	5 (63%)	12 (55%)	
Parietal Lobe	5 (4.4%)	1 (9.1%)	1 (13%)	3 (14%)	
Temporal Lobe	14 (12%)	6 (55%)	2 (25%)	6 (27%)	
Thalamus	1 (0.9%)	0 (0%)	0 (0%)	1 (4.5%)	
Brain, NOS (Unknown)	73 (64%)	14	38	21	
**Median OS ^3^**	14.1 [11.7–16.7]	13.3 [11.9–28.7]	14.3 [8.0–17.5]	12.6 [10.9–19.9]	

^1^ Continuous variables are displayed as average ± SD and categorical variables are displayed as sample n (%). ^2^ *p*-value determined for continuous variables by Kruskal–Wallis rank sum test and by *χ*^2^-test or Fisher’s Exact test for categorical variables excluding unknown category. ^3^ Median OS (overall survival) displayed as median [95% confidence interval].

**Table 2 cancers-16-04256-t002:** Summary of upregulated genes identified after correction for multiple testing. All significant genes were identified in Cluster 1 of radiomic consensus clusters.

Symbol	Chr	Description	log_2_FC ^1^	*p*	*p*-Adj.
*C5orf46*	5	chromosome 5 open reading frame 46	1.65	6.80 × 10^-5^	0.04
*H3C8*	6	H3 clustered histone 8	1.23	6.50 × 10^-5^	0.04
*LOC124904616*	1	RNA, Variant U1 Small Nuclear 28 (ENSG00000277918)	1.16	3.60 × 10^-5^	0.033
*SMYD2*	1	SET and MYND domain containing 2	0.82	1.10 × 10^-4^	0.049
*RPL39*	X	ribosomal protein L39	0.65	1.10 × 10^-4^	0.049
*TRIP4*	15	thyroid hormone receptor interactor 4	0.64	2.00 × 10^-6^	0.021
*KRCC1*	2	lysine rich coiled-coil 1	0.57	7.20 × 10^-6^	0.023
*BUD31*	7	BUD31 homolog	0.53	1.70 × 10^-5^	0.026
*TFB2M*	1	transcription factor B2, mitochondrial	0.51	5.70 × 10^-6^	0.023

^1^ log_2_FC = log_2_ fold-change of gene expression.

**Table 3 cancers-16-04256-t003:** Summary of downregulated genes identified after correction for multiple testing. All significant genes were identified in Cluster 1 of radiomic consensus clusters.

Symbol	Chr	Description	log_2_FC ^1^	*p*	*p*-Adj.
*SLC25A42*	19	solute carrier family 25 member 42	−0.53	2.30 × 10^-5^	0.031
*CASKIN2*	17	CASK interacting protein 2	−0.54	5.70 × 10^-5^	0.038
*CNNM2*	10	cyclin and CBS domain divalent metal cation transport mediator 2	−0.59	3.70 × 10^-5^	0.033
*ZBED3*	5	zinc finger BED-type containing 3	−0.60	5.20 × 10^-5^	0.037
*ADCY9*	16	adenylate cyclase 9	−0.64	1.30 × 10^-5^	0.025
*GRIK5*	19	glutamate ionotropic receptor kainate type subunit 5	−0.66	5.20 × 10^-5^	0.037
*ST3GAL5*	2	ST3 beta-galactoside alpha-2,3-sialyltransferase 5	−0.66	9.70 × 10^-6^	0.024
*PC*	11	pyruvate carboxylase	−0.68	6.30 × 10^-5^	0.04
*CNKSR3*	6	CNKSR family member 3	−0.71	7.80 × 10^-5^	0.042
*SORBS1*	10	sorbin and SH3 domain containing 1	−0.72	3.50 × 10^-5^	0.033
*DLG5*	10	Discs Large MAGUK Scaffold Protein 5	−0.78	4.00 × 10^-5^	0.033
*F10*	13	coagulation factor X	−0.81	1.40 × 10^-5^	0.025
*RNF39*	6	Ring Finger Protein 39	−0.84	3.50 × 10^-5^	0.033
*PRX*	19	periaxin	−0.85	5.40 × 10^-5^	0.037
*CRACD*	4	capping protein inhibiting regulator of actin dynamics	−0.88	2.90 × 10^-5^	0.033
*ATOH8*	2	atonal bHLH transcription factor 8	−1.03	6.40 × 10^-6^	0.023
*ARHGEF4*	2	Rho guanine nucleotide exchange factor 4	−1.08	8.10 × 10^-6^	0.023
*NAT8L*	4	N-acetyltransferase 8 like	−1.11	3.70 × 10^-5^	0.033
*ADAMTS8*	11	ADAM metallopeptidase with thrombospondin type 1 motif 8	−1.16	5.10 × 10^-5^	0.037

^1^ log_2_FC = log_2_ fold-change of gene expression.

## Data Availability

This study utilized data from three publicly available datasets: The Cancer Genome Atlas (TCGA) PanCancer Atlas glioblastoma (TCGA-GBM) dataset, brain lower grade glioma (TCGA-LGG) dataset, and the Clinical Proteomic Tumor Analysis Consortium (CPTAC) GBM (CPTAC-GBM) dataset.

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
