# Peer review of "Radiomic Consensus Clustering in Glioblastoma and Association with Gene Expression Profiles"

_cancers, 2024, doi:10.3390/cancers16244256_

Round 1
Reviewer 1 Report
Comments and Suggestions for Authors
In this research work, authors attempt to analyse the relationship between radiomic features from MRI and gene expression profiles in glioblastoma. The study has however some deficiencies.
1. It would be helpful for readers of the paper who are unfamiliar with the radiomic feature extraction step if a flowchart could be included for the Radiomic Feature Extraction section.
2.In radiomics, certain features may exhibit a strong correlation, however they may also encompass diverse aspects of tumor heterogeneity. Thus it may be helpful to carry out further analysis to avoid overriding important features due to excessive correlation.
3. For removing features with no variance,it may be helpful to state the threshold.
4. A more nuanced discussion of potential clinical applications and therapeutic implications is needed.
5. What is worrisome is the fact that identified clusters did not show any inter cluster differences with respect to patient demographics and total survival time which raises concerns on the clinical significance as well as discriminatory ability of the identified radiomic clusters.
Author Response
Comment 1: It would be helpful for readers of the paper who are unfamiliar with the radiomic feature extraction step if a flowchart could be included for the Radiomic Feature Extraction section.
Response 1: Thank you for your suggestion; we have included a flowchart in the Radiomic Feature Extraction section of the Methods (Figure 1, line 136). Furthermore, the review published by Singh and colleagues (2021) provides a comprehensive overview of the general steps involved in radiomic feature extraction as well as the utility of radiomics in glioma research. For readers who may be less familiar with radiomics, we have added in this reference for further information (Citation #12, https://doi.org/10.1038/s41416-021-01387-w).
Comment 2: In radiomics, certain features may exhibit a strong correlation, however they may also encompass diverse aspects of tumor heterogeneity. Thus, it may be helpful to carry out further analysis to avoid overriding important features due to excessive correlation.
Response 2: Thank you for your feedback; we agree that certain correlated features may encompass unique aspects of tumor heterogeneity but also acknowledge that including additional features that are highly correlated increases the risk of over-inflating cluster stability. In order to assess cluster stability across different correlation thresholds, we performed additional sensitivity analyses with correlation cut-offs at 0.9, 0.99, and 0.999 and assessed similarity with polychoric correlation. Cluster assignment was strongly correlated (>0.9) regardless of selected threshold. For downstream analyses assessing differential gene expression, we chose 0.95 as a strict and commonly used absolute correlation cutoff to avoid excluding potentially meaningful features while mitigating the risk of cluster stability over-inflation. We have added the results of the polychoric correlation analysis to the text (p4. line 143) and to the Supplemental Materials (Figure S1).
Comment 3: For removing features with no variance,it may be helpful to state the threshold.
Response 3: No variance was considered as variance=0. This has been included in the manuscript (p.4, line 141).
Comment 4: A more nuanced discussion of potential clinical applications and therapeutic implications is needed.
Response 4: Thank you for your feedback; we have provided several revisions to the Discussion and Conclusion sections of the manuscript to include more information about clinical and therapeutic implications of this study. We have included the following text: “Together, these findings suggest that the heterogeneity captured by radiomic consensus clustering may represent molecular differences across GBM that are independent of pa-tient demographics but may hold promise in the future for targeted therapies.” (p.9, line 328) and “Non-invasively identifying unique molecular profiles across a cohort of patients with GBM may provide further insight into the mechanisms of drug-resistance in tumors and provide additional information for selecting targeted therapies for GBM” (p.11, line 425).
Comment 5: What is worrisome is the fact that identified clusters did not show any inter cluster differences with respect to patient demographics and total survival time which raises concerns on the clinical significance as well as discriminatory ability of the identified radiomic clusters.
Response 5: Thank you for your feedback. The lack of association between inter-cluster differences across patient demographics and total survival may represent intra-tumor molecular heterogeneity which is captured and clustered with radiomic features and demonstrated through unique molecular profiles. Without targeted therapies for the treatment of GBM, it may be unlikely for survival to correspond with these captured differences. This study aims to highlight the role of radiomics and consensus clustering in stratifying GBM by unique molecular profiles; an approach that could be used to guide targeted therapies in the future.
Reviewer 2 Report
Comments and Suggestions for Authors
Comments on Manuscript ID cancers-3352614
Title Radiomic Consensus Clustering in Glioblastoma and Association with Gene Expression Profiles
Authors Wroblewsk et al
Comments
This is an interesting and unique study of gene expression profile in GBM patients to evaluate the differential expression pathway analysis by applying gene set enrichment approach of GBM patients with their clinical parameters. There are some concerns need to clarify:
1. In the selection of patient’s population, the author stated that the average age limit was 60 years, Cam author provide the information about inclusion of the number of pediatric or teen-age patients.
2. What was the criteria or reason to include 80-90% white population.
3. The author reported the vital status of the patient included in this study and the survival rate is 3-20% at the time of the study and can the author provide the data that whether the mortality happened during the study period and what was the average survival timeline of these selected patient population.
4. Can the author provide the clinical basis of a very high percentile of mortality of these patients.
5. This study has been categories the gene expression profile into three clusters. Is it possible to categories these clusters based on alive and deceased subjects which will be further helpful identified the signature upregulate and down regulate genes profile and pathway analysis between survival and mortality group.
6. In table 2 the authors provide the list of the differential expressions (FDR based) of up regulated and down regulated genes which should be separate into two table up regulate and down regulated gene sets.
7. In figure-1, the author provided the number of down and upregulated genes under two Classifications nominal and FDR. The FDR shows only in Cluster-1, Since the Nominal DEG genes listed in all of three clusters and can the author provide the common genes in more than two clusters and presence in all of the three clusters.
8. From the gene expression profile in both format DEG and pathway analysis approach, is it possible to the author to provide the most important genetic profile responsible for the least and most matching metrics of survival and mortality process in the selected patients of this study.
Author Response
Comment 1: In the selection of patient’s population, the author stated that the average age limit was 60 years,Cam author provide the information about inclusion of the number of pediatric or teen-age patients.
Response 1: No pediatric or teen-age patients were included in this study. Due to the data inclusion requirements and available data from the TCGA and CPTAC-3 databases, this study was limited to adult patients with GBM. This information has been added to the Methods section (p.4, line 111).
Comment 2: What was the criteria or reason to include 80-90% white population.
Response 2: The high proportion of patients of White racial identity in our cohort represents the limitation of available data. Considering our inclusion criteria for this study: patients with a GBM diagnosis along with availability of both multi-parametric MRI data and clinical information from the TCGA-GBM, TCGA-LGG, or CPTAC-GBM databases, we were limited. In the future, we hope additional data availability will enable analysis of a more diverse cohort.
Comment 3: The author reported the vital status of the patient included in this study and the survival rate is 3-20% at the time of the study and can the author provide the data that whether the mortality happened during the study period and what was the average survival timeline of these selected patient population.
Response 3: Methods utilized for survival analysis have been added to the Methods section of the manuscript (2.6 Statistical Analyses, p.5). Furthermore, the median overall survival of the selected patient cohort was 14.1 [95% CI = 11.7-16.7] months; this metric has been included in the Results section of the manuscript (p.6, line 259). We have added the following text to the manuscript: “Overall survival (OS) was assessed with Kaplan-Meier curves with survival time considered as “days to death” for deceased patients and “days to last follow-up” for patients reported living. Differences in overall survival curves across radiomic clusters was assessed using log-rank test. Median OS was calculated based on the Kaplan-Meier curve for the entire cohort and displayed as median [95% confidence interval].”
Comment 4: Can the author provide the clinical basis of a very high percentile of mortality of these patients.
Response 4: Thank you for your feedback. The present cohort of participants with GBM was selected based on data availability considerations (i.e., availability of both multi-parametric MRI data and clinical information) rather than clinical selection. Notwithstanding, we acknowledge that the proportion of deceased participants in our study (86%) is somewhat higher than previously reported in the full TCGA-GBM (76.8%) database. Given that alive status represents the time to last follow-up, which ranged from 0 days to over 7 years, we feel that vital status does not represent a meaningful statistic for this study. Instead, we have now reported the median overall survival of our cohort (14.1 months [95%CI=11.7-16.7), which is comparable to those previously reported in the full TCGA-GBM (13.9 months) database (https://doi.org/10.1016/j.cell.2013.09.034).
Comment 5: This study has been categories the gene expression profile into three clusters. Is it possible to categories these clusters based on alive and deceased subjects which will be further helpful identified the signature upregulate and down regulate genes profile and pathway analysis between survival and mortality group.
Response 5: We agree that stratifying each cluster by alive/deceased status would help to further identify differential gene expression within clusters, and we hope these methods can be expanded to larger cohorts in the future. As mentioned above, vital status may not serve as an appropriate measure for comparison in this study given that the time of last follow-up ranged from 0 days to over 7 years, so this has been replaced with median overall survival. Furthermore, for the present study we are limited by the practical consideration of sample size, this study included only 16 patients who were reported living at last follow-up, which prevents additional appropriately powered sub-group analyses.
Comment 6: In table 2 the authors provide the list of the differential expressions (FDR based) of up regulated and down regulated genes which should be separate into two table up regulate and down regulated gene sets.
Response 6: Thank you for this feedback; Table 2 has been separated into two tables: Table 2 now presenting up-regulated genes and Table 3 representing down-regulated genes.
Comment 7: In figure-1, the author provided the number of down and upregulated genes under two Classifications nominal and FDR. The FDR shows only in Cluster-1, Since the Nominal DEG genes listed in all of three clusters and can the author provide the common genes in more than two clusters and presence in all of the three clusters.
Response 7: We have added a table to the Supplementary materials that describes nominally significant and significant-after testing for multiple corrections genes that are common across more than one cluster (Table S1), which is referenced in the text on line 294.
Comment 8: From the gene expression profile in both format DEG and pathway analysis approach, is it possible to the author to provide the most important genetic profile responsible for the least and most matching metrics of survival and mortality process in the selected patients of this study.
Response 8: Thank you for your suggestion. This study was aimed to highlight the potential utility of radiomic consensus clustering to identify unique molecular profiles in GBM; therefore, we believe that identifying genetic profiles solely across survival status is outside the scope of this study and has been assessed elsewhere in the TCGA-GBM cohort (e.g., https://doi.org/10.1016/j.cmpbup.2022.100051). Furthermore, we agree that stratifying by survival status would provide further insight into the gene expression profiles within each consensus cluster; however, the limited sample size in this study prevents additional appropriately powered sub-group analyses.
Reviewer 3 Report
Comments and Suggestions for Authors
The article “Radiomic Consensus Clustering in Glioblastoma and Association with Gene Expression Profiles” is very interesting.
I will briefly make some points to improve the methodology.
- List the statistical tests that will be performed with the demographic data and their significance value. They are subsequently commented on in the results section.
- Mention the statistical method used to determine the probability of survival.
Please review the data in Table 1, "Other or Unknown" 10?, "Ethnicity (Hispanic/Not Hispanic"??
Add recent publication on radiomic glioblastoma
Thank you very much
Author Response
Comment 1: List the statistical tests that will be performed with the demographic data and their significance value. They are subsequently commented on in the results section.
Response 1: Statistical tests that were performed to assess demographic and clinical characteristic differences across radiomic clusters have been added to the Methods section (2.6 Statistical Analyses, p.5). Thank you for bringing this to our attention. The following description has been added to the manuscript: “Demographic variables and clinical characteristics were compared across radiomic clusters using Kruskal-Wallis rank sum test for continuous variables and χ2 test or Fisher’s Exact test, when appropriate, for categorical variables. Continuous variables were dis-played as the mean +/-standard deviation (SD) and categorical variables were displayed as the sample n (percent, %).”
Comment 2: Mention the statistical method used to determine the probability of survival.
Response 2: Statistical methods used to in survival analysis have been added to the Methods section (2.6 Statistical Analyses, p.5) of the manuscript: “Overall survival (OS) was assessed with Kaplan-Meier curves with survival time considered as “days to death” for deceased patients and “days to last follow-up” for patients reported living. Differences in overall survival curves across radiomic clusters was assessed using log-rank test. Median OS was calculated based on the Kaplan-Meier curve for the entire cohort and displayed as median [95% confidence interval].”
Comment 3: Please review the data in Table 1, "Other or Unknown" 10?, "Ethnicity (Hispanic/Not Hispanic"??
Response 3: Thank you for pointing this out. “Other” was erroneously included; Table 1 has now been revised to “Hispanic or Latino”, “Not Hispanic or Latino”, and “Unknown” for the Ethnicity section.
Comment 4: Add recent publication on radiomic glioblastoma
Response 4: Thank you, the following recent radiomic glioblastoma publications have been cited: (A) Citation #14: https://doi.org/10.1016/j.currproblcancer.2024.101156; (B) Citation #15: https://doi.org/10.1038/s41598-024-78189-6.
Reviewer 4 Report
Comments and Suggestions for Authors
In cancers-3352614, Wroblewski et al examine the association between Radiomic Consensus Clustering in Glioblastoma and Gene Expression Profiles. The topic pf this bioinformatic study is interesting and fits well the scope of Cancers. The reviewer has no objection to pass this manuscript.
(1) The limitation of this study is obvious. The sample size is very small. Dose gender, age and race have impact on gene expression?
(2) What kind of samples were analyzed for the gene expression? The gene expression profile is also tissue specific.
Author Response
Comment 1: The limitation of this study is obvious. The sample size is very small. Dose gender, age and race have impact on gene expression?
Response 1: Thank you for your comments and question; from VariancePartition analysis (Figure S3), after batch correction, we observe the variance explained by age and sex covariates (0.75%, 0.54%, respectively) represents a nominal contribution on gene expression after adjusting for radiomic clusters. Race and ethnicity were strongly correlated with dataset source (Figure S2) and were therefore not assessed as covariates in downstream analyses.
Comment 2: What kind of samples were analyzed for the gene expression? The gene expression profile is also tissue specific.
Response 2: Primary GBM were the samples analyzed for the gene expression data. Level 3 gene expression data was downloaded from the GDC data portal (https://portal.gdc.cancer.gov) originating from the TCGA-GBM, TCGA-LGG and CPTAC-GBM databases. We have clarified this in the methods section (p.5, line 177): “Gene expression data from primary GBM tumor samples was available for 69 out of the 114 patients included in this study…”.